# Recent Advances in Precision Diamond Wheel Dicing Technology

**DOI:** 10.3390/mi16101188

**Published:** 2025-10-21

**Authors:** Fengjun Chen, Meiling Du, Ming Feng, Rui Bao, Lu Jing, Qiu Hong, Linwei Xiao, Jian Liu

**Affiliations:** 1School of Robot Engineering, Wenzhou University of Technology, Wenzhou 325000, China; dumeiling@stu.wzu.edu.cn (M.D.); 20240047@wzut.edu.cn (L.J.); 20240050@wzut.edu.cn (Q.H.); 20240062@wzut.edu.cn (L.X.); 20220048@wzut.edu.cn (J.L.); 2School of Mechanical and Electrical Engineering, Wenzhou University, Wenzhou 325035, China; fming@wzu.edu.cn; 3College of Mechanical and Vehicle Engineering, Hunan University, Changsha 410082, China; baorui@hnu.edu.cn

**Keywords:** dicing process, diamond blade, control algorithms, tool wear, process parameters

## Abstract

Precision dicing with diamond wheels is a key technology in semiconductor dicing, integrated circuit manufacturing, aerospace, and other fields, owing to its high precision, high efficiency, and broad material applicability. As a critical processing stage, a comprehensive analysis of dicing technologies is essential for improving the machining quality of hard-and-brittle optoelectronic materials. This paper reviews the core principles of precision diamond wheel dicing, including dicing processes and blade preparation methods. Specifically, it examines the dicing mechanisms of composite and multi-mode dicing processes, demonstrating their efficacy in reducing defects inherent to single-mode approaches. The review also examines diverse preparation methods for dicing blades, such as metal binder sintering and roll forming. Furthermore, the roles of machine vision and servo control systems are detailed, illustrating how advanced algorithms facilitate precise feature recognition and scribe line control. A systematic analysis of key components in grinding wheel dicer is also conducted to reduce dicing deviation. Additionally, the review introduces models for tool wear detection and discusses material removal mechanisms. The influence of critical process parameters—such as spindle speed, feed rate, and dicing depth—on dicing quality and kerf width is also analyzed. Finally, the paper outlines future prospects and provides recommendations for advancing key technologies in precision dicing, offering a valuable reference for subsequent research.

## 1. Introduction

Semiconductor materials are indispensable foundations of modern technology, playing critical roles in fields from integrated circuits and optoelectronics to aerospace and defense [1,2]. Given their high hardness, semiconductor materials are typically cut using diamond grinding wheel blades. Materials diced by diamond grinding wheel blades primarily include monocrystalline silicon, ceramics, gemstones, optical glass, and gallium arsenide (GaAs). These materials are often selected for their superior physical and optical properties [3]. However, these very properties also render the materials prone to damage and cracking during dicing, which can compromise the quality of the finished workpiece. Mitigating these defects remains a persistent challenge in current research on diamond grinding wheel dicing technology.

Ultra-thin diamond grinding wheels are employed for dicing various workpieces, such as monocrystalline silicon, enabling precise control over the blade’s dicing depth. Characterized by low dicing force, thin kerf width, and low operational costs, diamond wheel dicing remains the predominant dicing technique in industry. Recent research has prioritized dicing quality enhancement to meet the demands of “high precision and high efficiency.” Studies on diamond wheel dicing have revealed that different dicing processes result in varying degrees of workpiece damage.

To advance diamond wheel dicing technology, this study focuses on key precision techniques. It addresses the dicing process methodology, blade preparation, and machine control systems, incorporating specific optimizations based on visual and servo control principles in Figure 1. Additionally, the paper details the coolant management system of the dicing machine, post-operation truing and dressing procedures, the blade wear prediction model during dicing, and the workpiece material removal mechanism. These elements, together with experimental and simulation data on dicing parameters and their optimizations, are integrated and analyzed from diverse control perspectives to develop multiple control strategies. By systematically synthesizing these diverse topics, this review establishes a foundational framework for future advancements in semiconductor dicing technology. Finally, the study discusses the limitations of the current work and outlines promising future research directions for key grinding wheel dicing technologies.

## 2. Semiconductor Dicing Processes

### 2.1. Multi-Mode Dicing Processes

The choice of dicing method is critical to final die quality. Consequently, researchers have categorized dicing strategies primarily by the number of passes per operation or by specific grinding wheel parameters. Shen et al. [4] presented a single-step dicing process. Through experimental analysis of the contact arc angle between the grinding wheel and workpiece in Figure 2a, the research showed that the optimal arc angle is 20°, The dicing force tends to “plow” through the material rather than shatter it, thereby reducing crack propagation and resulting in improved dicing performance. To overcome the low yield of traditional metal-bonded diamond blades, which leads to grain dislodgement or brittle fracture and impedes precise straight dicing of SiC substrates, Fujita et al. [5] developed a novel ductile-mode dicing process, as illustrated in Figure 2a. The process utilizes polycrystalline diamond blades with a thickness of only 50 μm. Ductile-mode machining is achieved through a sequence of steps: grinding the blade periphery, using wire-EDM to remove the central material, and applying specialized EDM to reduce thickness. Crucially, this method prevents metal film adhesion to the dicing edge. This blade operates at high speed with sufficiently low vibration and removal volume to shift the material removal mechanism from brittle fracture to plastic flow, effectively preventing both chip adhesion and loss of dicing sharpness. Chen et al. [6] investigated the dual-wheel dicing process depicted in Figure 2b. The first wheel features a smaller abrasive grain size, greater thickness, and a V-shaped dicing edge, while the second wheel is thinner with a larger grain size. After the first wheel dices to a specific depth, the second wheel completes the separation, reportedly increasing workpiece strength by 15%. Yin et al. [7] proposed a layered dicing process for silicon wafers, as shown in Figure 2c. Compared to single-pass dicing, the layered dicing process uses a relatively shallow dicing depth in the first pass, scribing down to half the UV tape thickness. The process reduces cutter stress and wear, thereby minimizing scratch chipping and kerf size.

### 2.2. Composite Dicing Process

Abrasive wheel composite dicing is an advanced dicing method that marries the principles of abrasive wheel dicing with laser technology. By optimizing all critical parameters, this approach capitalizes on the grinding wheel’s high efficiency and the laser’s exceptional precision to achieve the required dicing quality and meet technical specifications. This process is commonly used for dicing semiconductor materials. To address wafer damage during semiconductor dicing, Lee et al. [8] implemented this hybrid approach as depicted in Figure 3a. The process begins with laser-induced groove formation along the dicing centerline on the wafer surface. Next, the wafer is inverted, and finally, a precision saw blade is inserted into the opposite side of the laser-initiated grooves for separation. This methodology eliminates brittle fracture at the wafer edges while producing groove tips of minimal curvature and a high aspect ratio, thereby outperforming conventional groove geometries. Device failure induced by wafer dicing remains a critical technological challenge in semiconductor advancement. Conventional cutting processes fail to preserve surface pattern integrity or prevent chipping during wafer separation. Addressing the ductility of silver layers and post-dicing edge irregularity, Li et al. [9] implemented a laser-grinding wheel hybrid technique for front and back-side silicon wafer processing. To eliminate silver layer curling, Li utilized complete laser ablation of all silver layers, achieving precision grooves 45 μm in width and 5 μm in depth. This approach achieved uniformly straight edges and consistent dimensions on both surfaces of the separated dies.

Li and Lee both utilize hybrid methodologies integrating traditional grinding wheel dicing with laser technology, yet address distinct target issues through divergent experimental approaches. Li’s research prioritizes the optimization of process parameters, including laser energy and grinding wheel specifications. In contrast, Lee’s work targets stress concentration and fracture mechanics within hybrid dicing processes, thereby mitigating the inherent limitations of single-mode methods.

To address the challenges posed by 4H-SiC wafers including extreme material hardness, interfacial failure in alloy backing layers, and performance degradation under suboptimal thick-section dicing, Qu et al. [10] developed two advanced hybrid techniques validated in Figure 3b: laser ablation-stealth dicing (LAS) and micro-grinding-implicit dicing (MGID). The LAS technique resolves intermediate alloy tearing via precision laser ablation. Conversely, by ensuring compatibility, precise strain control, and minimized mechanical stress, the MGID process successfully maintains epitaxial layer integrity and eliminates backlayer-SiC separation.

The laser-grinding wheel hybrid dicing process synergistically integrates the respective advantages of laser ablation and mechanical dicing, effectively circumventing the inherent limitations of individual methods. Through optimized process sequencing, this approach mitigates product quality degradation caused by single-mode defects while reducing operational expenditures. Consequently, these combined benefits position the hybrid dicing process as a highly effective method for industrial semiconductor applications. Overall, the dicing process presents different technological paths for the challenges of processing hard and brittle materials. From single dicing process to innovative tool technology, to multi-mode process and multi job combination, from pursuing macro machining quality to ultimate sub surface quality, which demonstrates the diversity and innovation of dicing technology. In practical applications, the choice of process depends on the characteristics of the material, especially the higher requirements for workpiece strength and production cost-effectiveness. However, the potential of the dicing process provides more references for the subsequent processing of superhard substrate materials.

## 3. Preparation of Ultra-Thin Dicing Blades

In industrial applications, the dicing performance of diamond ultra-thin grinding wheels varies significantly with the manufacturing method used. Common preparation techniques such as binder sintering, hot pressing, electroplating, and brazing each offer distinct advantages for specific materials.

### 3.1. Preparation of Dicing Blades by Sintering

The development of modern ultra-precision diamond grinding wheels is propelled by two key factors: innovations in the bond system and, more fundamentally, advances in high-end synthetic diamond abrasives. These include chemical vapor deposition (CVD) microcrystalline diamonds and high-strength, high-grade high-pressure high-temperature (HPHT) single-crystal diamonds. Such tailored abrasive sources provide the material basis for uniform grain distribution, high retention strength, and consistent wear behavior, establishing them as a core driver behind the trend toward ultra-thin, high-precision dicing technology. In the sintering method, synthetic diamond or CBN abrasives are blended with metallic powders in precise ratios and then sintered, resulting in ultra-thin dicing blades with exacting geometries. Wang et al. [11] uniformly blended copper-coated graphite with conventional graphite according to predetermined ratios in their experimental design. A φ58 mm × φ40 mm × 0.5 mm test grinding wheel was fabricated from an 80 wt% Cu-20 wt% Sn composition by mixing and cold-pressing the powder into a green compact, followed by resistance sintering. The incorporation of copper-coated graphite enhanced the hardness and flexural strength of the metal bond matrix by up to 8.4% and 25.5%, respectively. Sumiya et al. [12] prepared grinding wheels with a diameter of 3.2 mm using nano polycrystalline diamond directly transformed and sintered under high pressure and high temperature to study the grinding performance of single crystal diamond grinding wheels. Xiao et al. [13] utilized metal bonds to produce ultra-thin dicing blades, as illustrated in Figure 4a. They developed a micro-sintering model for metal-bonded diamond grinding wheels based on fluid dynamics and the discrete element method. At identical intermediate and final sintering temperatures, octahedral diamonds exhibited lower thermal stress and superior dicing performance compared to dodecahedral diamonds.

Ultra-thin dicing blades produced by sintering are generally characterized by high bond strength, excellent wear resistance, and a long service life. However, a potential drawback is the risk of non-uniform abrasive distribution. He et al. [14] utilized FDMS technology to fabricate ultra-thin diamond blades. The process involved sequential steps of mixing, filament collection, printing, debinding, and sintering to produce the filaments. The FDMS-based manufacturing approach further enabled blade fabrication through printing, degumming, sintering, and precision grinding in the post-processing stage, as illustrated in Figure 4b. Feng et al. [15] proposed a roll-cutting forming method for manufacturing ultra-thin diamond dicing blades, as illustrated in Figure 4c. The blade fabrication process encompassed resonant acoustic mixing, tape casting, roll-cutting, laser slitting, debinding, and sintered surface polishing, enabling the continuous production of high-precision blades with thicknesses as low as 0.048 mm. Miao et al. [16] fabricated an ultra-thin grinding wheel with 75% ultra-high porosity, honeycomb structure, and homogeneous microstructure by combining gel-casting molds with pore-forming agents. This approach prevented diamond particle agglomeration and enhanced Si wafer machining accuracy. Zhang et al. [17] developed a grinding wheel using a novel glass-bond formulation incorporating nano-SiO_2_ and nano-Ce_2_, as shown in Figure 4d. Grinding experiments demonstrated the nano-glass-bond wheel’s superior performance over conventional counterparts. Wang et al. [18] employed a nanosecond pulsed laser to ablate a miniature electroplated diamond grinding wheel, revealing how laser parameters influence structure fabrication, which guided the successful preparation of a bionic phyllotactic grinding wheel.

**Figure 4 micromachines-16-01188-f004:**
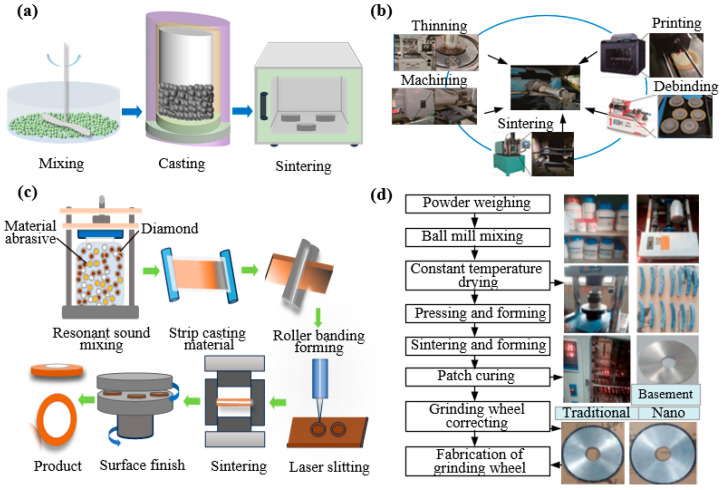
Four preparation methods for diamond grinding wheels: (**a**) Sintering process [13]. (**b**) Ceramic bond method [14]. (**c**) Rolling cutting method [15]. (**d**) Sintering grinding method [17].

In the process of fabricating ultra-thin diamond blades via sintering, dicing performance can be enhanced through stringent control of processing parameters, the incorporation of novel materials, precise regulation of sintering temperatures, and optimized porosity design. Diversifying preparation methods offers a pathway to enhance dicing performance and overcome the inherent constraints of sintering.

### 3.2. Hybrid Fabrication Methodology

The development of preparation processes for ultra-thin dicing blades continues to advance, with researchers exploring diverse fabrication methods to enhance performance. Kong et al. [19] produced diamond tools using a melt wire manufacturing method, as shown in Figure 5a. The process started with a raw material consisting of 55 vol% diamond powder and 45 vol% binder. This material was extruded into filaments using a single-screw extruder and then printed into corresponding tool shapes using an FDM printer. This method effectively reduced internal defects and improved the uniformity of diamond distribution. Zhang et al. [20] prepared Ni-W/diamond ultra-thin slices with different tungsten contents by using a self-developed cathode spin plating device and composite electrodeposition technology, as shown in Figure 5b. By modifying the nickel sulfate plating solution with different concentrations of sodium tungstate, they achieved a significant enhancement in blade performance. The improvements included a 47.9% lower deviation coefficient, a 30.6% reduction in chip size, and 31.6% less radial wear relative to the conventional Ni/diamond blade. Fujita et al. [21] produced a high-precision ultra-thin dicing blade with a thickness of 20 μm and a crack-free, high-precision ultra-thin dicing blade with a thickness of 20 μm from a polycrystalline diamond (PCD) blank using an electric spark machining process and UV-assisted polishing, as shown in Figure 5c. The experiments show that the blade prepared by this method can dice the ultra-fine groove with a width of 20 μm and a depth of 1 mm on the hard and brittle materials. Rendi Kurniawan et al. [22] developed an innovative ultrasonic dicing blade to address the pain point of edge collapse and cracking caused by traditional dicing processes. By establishing electromechanical impedance and harmonic analysis methods, they studied the characteristics of resonant frequency, anti-resonant frequency, and radial vibration amplitude. Through simulation analysis, the new blade can generate an amplitude of about 0.2–0.8 μm at a resonant frequency of 42–43 kHz, significantly improving the quality of microgroove dicing. Li et al. [23] used a new bonding method to manufacture a diamond blade taking a new Ti-15/75Al binary alloy as the bonding material. The blade has a high hardness and a large enough porosity, which is beneficial to dicing SiC materials. Through understanding the influence of sintering temperature on microstructure and friction mechanical properties of hard-brittle materials, Li also found that sufficient porosity and self-sharpening capacity play an important role in meeting the dicing quality of sapphire [24]. Owing to their complex structure, high porosity, and controllable pore size, porous ceramic-bonded grinding wheels are widely employed for high-efficiency, precision grinding of hard and brittle materials. Huang et al. [25] proposed to use direct ink writing technology to manufacture three kinds of grinding wheels, solid structure, triangle, and dot matrix structures. As shown in Figure 5d, the material removal rate of triangular structure grinding wheel and dot matrix structure grinding wheel is higher than that of solid structure grinding wheel.

High-quality ultra-thin diamond dicing blades deliver superior dicing performance owing to their exceptionally narrow kerf widths and high-precision machining capabilities. The higher manufacturing costs and process complexity are justified by superior benefits like efficient material removal and long service life, which can be further improved with novel materials and advanced processing to pave the way for new applications.

### 3.3. Tool Wear

Operational conditions and workpiece material properties greatly affect dicing blade lifespan, complicating the creation of precise adaptive prediction models. Wanyong Liang et al. [26] introduced nonlinear convergence factors and adaptive weighting factors to enhance the standard Golden Jackal Optimization (GJO) algorithm, developing an Adaptive Golden Jackal Optimization (AGJO)-based Gated Recurrent Unit (GRU) prediction model. As illustrated in Figure 6a, the proposed model achieves 1.96% higher accuracy in life prediction with 27.04% reduction in root mean square error compared to conventional methods. Shi et al. [27] established a tool life prediction framework using Adaptive Genetic Algorithm-optimized Backpropagation Neural Networks (AGA-BPNN) in Figure 6b. The methodology overcomes characteristic genetic algorithm drawbacks including slow convergence, significant fluctuations, and local optima trapping, ultimately attaining 91.76% predictive accuracy. While both models enhance predictive capability through algorithmic refinement, their applications diverge: AGJO-GRU excels in equipment lifespan forecasting, whereas AGA-BPNN proves superior for real-time quality monitoring and production efficiency optimization. Collectively, they demonstrate the critical role of optimization algorithms in advancing model performance. Maia et al. explored real-time carbide tool wear monitoring using Acoustic Emission (AE) and Short-Time Fourier Transform (STFT). STFT effectively identified key wear frequencies: 200–1000 kHz for abrasive wear and 350–550 kHz for crack propagation, enabling precise characterization of wear mechanisms [28].

Vertical forces during blade traversal can induce wafer edge chipping, and excessive edge annealing may jeopardize product integrity and safety. Shi et al. [29] enhanced cutting quality prediction by optimizing Bidirectional Long Short-Term Memory (BiLSTM) networks via an improved Sparrow Search Algorithm (SSA) in Figure 6c. This CLSSA-BiLSTM model addresses unidirectional propagation limitations in traditional neural networks, elevating R2 from 0.8875 to 0.934 with 22% reduced error metrics. Complementing the approach, Su et al. [30] implemented Backpropagation Neural Networks (BPNNs) for real-time tool wear detection using machine sensor data, achieving 75% higher prediction accuracy. Jeon et al. [31] developed a non-contact wear monitoring system leveraging edge diffraction effects in Figure 6d. Jeon analyzed interference patterns between transmission and diffracted waves at wheel edges, extracting abrasive wear signatures from diffraction patterns through cross-correlation, with findings notably showing that grinding wheel blades exhibit accelerated wear when processing hard-brittle materials. To enable real-time blade wear detection, Chen et al. [32] proposed a fiber-optic sensor-based method for grinding wheel blade wear monitoring, as illustrated in Figure 6e. Building upon this experimental foundation, Chen employed a Proportional-Integral-Derivative (PID) with feedforward and notch filter control algorithm, achieving a maximum blade wear measurement error of 3.8 μm after parameter optimization. Deng et al. [33] integrated bidirectional temporal residual neural networks with self-attention mechanisms and voting algorithms for blade damage detection, attaining accuracy exceeding 99.5% in testing.

Beyond real-time dicing quality prediction, tool condition during machining plays a key role in determining outcomes. To model the complex dynamics of tool dicing, a lateral vibration differential equation was established, and a multi-domain feature–based AI algorithm was used for blade condition prediction, resulting in a GA-optimized Backpropagation Neural Network (GA-BPNN) hybrid model [34]. To more clearly visualize the state of the dicing tool during the dicing process, He et al. [35] developed dicing and process models for ultra-thin diamond blades by integrating scripting with finite element analysis software. The results indicate that the optimal dicing quality for SiC wafers is achieved at a dicing depth of 6 μm. In tool wear, it is essential to select appropriate dicing prediction methods. Therefore, a summary of methods for predicting tool wear and dicing quality is indispensable, as shown in Table 1.

**Figure 6 micromachines-16-01188-f006:**
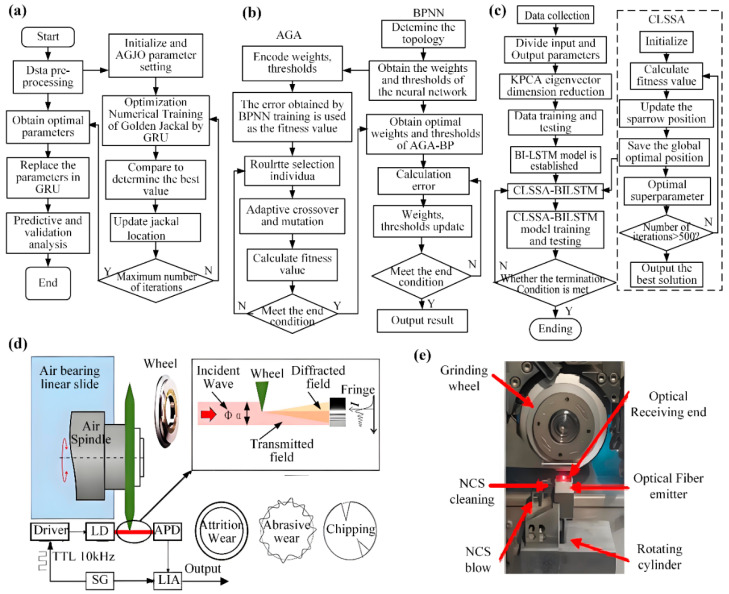
Models and detection principles for diced quality assessment: (**a**) Life prediction model [26]. (**b**) Neural network model algorithm [27,28]. (**c**) Dicing quality model [29]. (**d**) Edge diffraction principle [31]. (**e**) Optical fiber detection principle [32].

### 3.4. Grinding Wheel Dicing Trimming

In the dicing process, variations in dicing forces across different contact zones cause differential wear of the grinding wheel blade, resulting in a gradual deviation from circularity in the diamond-abrasive surface. Ensuring optimal profile geometry and precision in grinding wheel dicing blades makes effective wheel truing a critical challenge in the dicing industry and a key research focus globally. Current finishing methodologies, as shown in Figure 7, include mechanical, electrolytic, and hybrid techniques [36]. Finishing diamond wheels with high abrasive hardness remains particularly difficult [37]. Zhou et al. [38] proposed roll-grinding dressing using silicon carbide wheels to achieve efficient and precise truing of arc-profile diamond wheels for aspherical parallel grinding. By examining the geometric relationship between the grinding and dressing wheels, Zhou developed a mathematical model of the dressing path and theoretically assessed dressing efficiency and accuracy. Experimental validation showed peak-to-valley (PV) shape errors of less than 4 μm, alongside improved processing efficiency. Karpuschewski, B. et al. [39] who often employed dressing disks for grinding wheel dressing, proposed that the normal dressing force, which is responsible for the collapse of surface abrasive particles, accounts for the major kinematic model discrepancies observed during rotary dressing.

Chen et al. [40] proposed a vision-based truing system addressing low efficiency, insufficient accuracy, and harsh environments in diamond grinding wheel finishing. An optical imaging module, equipped with a parallel light source, dual telecentric lenses, and a CCD camera, provides the system with real-time data on wheel contour deviation. The data is transmitted via Siemens OPC UA protocol to an 828D CNC system, which dynamically adjusts the silicon carbide dressing wheel through multi-axis linkage control with high-precision servo and torque motors. Experimental results demonstrate contour truing errors below 0.01 mm with 32% higher efficiency than conventional methods. Chen et al. [41] developed a “Coaxial Rotating Electro-Refining Regeneration” process for regenerating electroformed diamond blades through controlled dissolution and thinning. Under an optimized cathode configuration and electrochemical conditions, the blades undergo slow rotation within the electrolytic cell, ensuring a gradual and uniform dissolution of the nickel matrix. The process allowed trapped grinding debris to escape while exposing fresh diamond abrasives, facilitating rapid blade recovery without mechanical stress or geometric distortion. Dai et al. [42] applied electrical discharge grinding (EDG) to dress bronze-bonded diamond wheels, analyzing the removal mechanism and modeling the discharge process in 3D. Via simulation, they determined the temperature fields to optimize grain cutting depth. The method efficiently discharges non-conductive diamonds, yet is mainly applicable to form-grinding wheels with coarser, irregular grains.

Furthermore, Watanabe et al. [43] identified that radial run-out of dicing tools induced chipping and cracking on machined surfaces. To address this challenge, a metal-bonded conductive polycrystalline diamond blade was installed on a microtome spindle and processed via pure-water electro-discharge machining. This enabled chipping-free machining of SiC, as shown in Figure 7e. To optimize dressing parameters for grinding Ti-6Al-4V titanium alloy with SiC wheels, Mukhopadhyay et al. [44] employed identical 0.75-carat single-point diamond dressers to condition the wheels at feeds of 5, 10, 15, 20, and 25 μm under varied dressing modes. Empirical evaluation demonstrated that 20 μm was the optimal dressing feed.

**Figure 7 micromachines-16-01188-f007:**
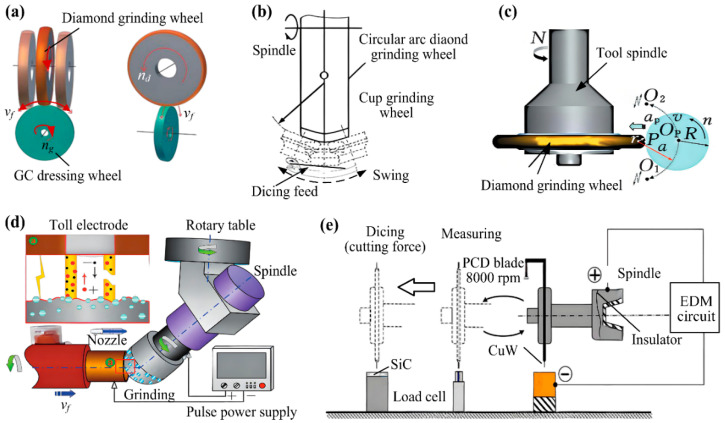
The mechanical dressing method [36]: (**a**) GC grinding wheel dressing. (**b**) Arc grinding wheel dressing. (**c**) Cup grinding wheel dressing. (**d**) Discharge dressing. (**e**) Electro-spark discharge trimming [43].

## 4. Dicing Control Method

### 4.1. Vision System Control

A high-precision visual positioning system ensures accurate dicing point alignment, reducing material waste. However, the measurement accuracy still poses a challenge for its widespread application. Therefore, a large number of researchers have improved the positioning accuracy of visual control from different technical levels. Khan, MF et al. [45] explored the possibility of actual and measured uncertainties in the calibration process of robot vision systems. Based on a series of experimental tests and computer simulations, error sensitivity analysis revealed that even small uncertainties can significantly affect the positioning accuracy of the vision system. Wang et al. [46] enhanced wafer image brightness through exponential transformation within a nonlinear image enhancement algorithm to achieve automated dicing path detection. In their method, template matching and an improved random sample consensus (RANSAC) algorithm localized chip center positions, improving image quality by 82% while restricting path-fitting errors to within 1.5 pixels. Wei et al. [47] integrated bilinear interpolation and sub-pixel edge detection algorithms on the Halcon platform, as shown in Figure 8b. On the basis of the original Canny edge coarse extraction, bilinear interpolation and sub-pixel methods are used to improve the accuracy of trajectory recognition, thereby further enhancing the dicing precision of the equipment. Compared to traditional Canny operators, this approach accurately identifies trajectories and adapts to high-precision dicing equipment, though it lacks next-trajectory planning capability. The method significantly enhanced dicing accuracy and production efficiency. By improving image quality and feature accuracy, the development of image preprocessing and edge detection methods also optimizes the tool motion trajectory, thereby signifying the visual positioning system’s advancement from mere perception to enhanced positional accuracy. To machine complex components like aircraft engine casings, Rodríguez et al. [48] proposed an automated vision-based method that uses blue light scanning to acquire 3D part data. By comparing this data with the theoretical model, the system adaptively generates machining paths to compensate for casting tolerances and ensure accurate edge location, providing a feasible strategy for precision machining parts with large geometrical deviations. Rico et al. [49] achieved 13 μm-level positioning accuracy through a non-contact wafer precision positioning system based on aerostatic bearing technology. In this system, controlled airflow generates viscous traction forces for wafer manipulation, while CCD sensors perform non-contact position measurement. The non-contact method prevents accuracy loss from workpiece wear debris and protects surfaces from damage, which is critical in semiconductor dicing. Chen et al. [50] proposed a visual positioning algorithm for semiconductor wafers that combines grayscale and line features, which uses an image pyramid and a hierarchical search strategy to accelerate feature detection by performing a rapid coarse search at low resolution and progressively narrowing the search area in higher resolutions, thereby reducing computational load and, with initial angle optimization, dicing down search time. Their improved template matching algorithm, which incorporated initial-angle optimization, reduced processing time to <80 ms, which is one quarter of the time required by conventional normalized cross-correlation algorithms in Figure 8a, while maintaining 99.25% matching success under extreme conditions.

**Figure 8 micromachines-16-01188-f008:**
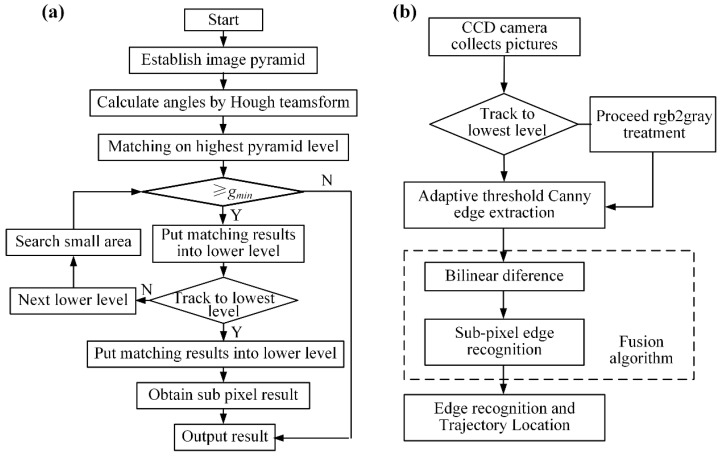
Flowchart of visual matching algorithms: (**a**) Improved template matching algorithm [50]. (**b**) Bilinear interpolation and sub-pixel edge recognition algorithm [51].

The visual positioning system (VPS) is the core component of intelligent grinding wheel dicing machines, enabling precise workpiece localization by acquiring and processing visual data. This functionality is fundamentally dependent on integrated sensor systems. Sun et al. [51] proposed an optimization methodology combining the Effective Independence Method, QR Decomposition, Modal Assurance Criterion, and Shannon Expansion Theorem for sensor placement in dicing machine vibration modal testing. Hammer impact modal testing verified the method’s effectiveness, with comparative analysis of pre-optimization modal data confirming its validity. The research moves beyond pure visual algorithms by integrating multiple optimization criteria. It dynamically demonstrates that visual positioning accuracy is primarily constrained by mechanical vibration, thereby addressing a key bottleneck in enhancing precision. The positioning and identification system critically governs the entire dicing process. Selecting optimal algorithms is paramount for enhancing dicing accuracy, as shown in Table 2. Naturally, the visual positioning system of a modern intelligent cutting machine constitutes a complex system integrating multiple technologies. The choice of a technological path is therefore dictated not only by general constraints like equipment accuracy and cost but also by the specific application’s priorities regarding positioning accuracy, speed, robustness, and overall cost. Furthermore, integration of the VPS with the central control module requires parallel advancements in intelligent efficiency and refinement of the control architecture.

### 4.2. Servo Control System

The servo control system of the dicing machine integrates a PLC, sensors, motor drivers, an operator panel, and related electrical components. The sensors monitor the machine’s motion states and workpiece positioning, transmitting the acquired data to the PLC for processing. Wu et al. [52] implemented a PID + feedforward 2DOF control algorithm to regulate servo motor current. This control architecture integrates a feedforward channel *G*_2_(*s*) and a PID feedback channel *G*_1_(*s*), with the system’s mathematical model derived from the block diagram in Figure 9a.(1)$$u=\left[G_{r}G_{y}\right]\left[\begin{array}{c} r\\ y \end{array}\right]=\left[\left(1+b\right)K_{p}+K_{I}\frac{1}{S}+\left(1+c\right)K_{D}sK_{p}+K_{I}\frac{1}{S}+K_{D}s\right]\left[\begin{array}{c} r\\ y \end{array}\right]$$

Fine-tuning PID + feedforward 2DOF parameters stabilizes the servo system, minimizes position error, and improves motion-axis tracking. An analysis of positional deviations under varying control parameters (proportional gain, integral time constant, velocity feed-forward gain) led to the development of contact and non-contact height measurement principles. These methods compensate for linear positioning errors during cutting, demonstrating the improved system’s strong potential for high-precision, high-speed semiconductor dicing. Liang et al. [53] developed a cascaded reduced-order linear active disturbance rejection controller that simultaneously compensates for system uncertainties and strengthens disturbance rejection capabilities. The controller employs a Gold Jackal Optimization algorithm for internal parameter optimization. As shown in Figure 9b, the controller employs a Gold Jackal Optimization algorithm for internal parameter optimization. This approach offers structural simplicity, minimal parameterization, and straightforward implementation to achieve high-precision servo axis control through synergistic integration. By incorporating the Golden Jackal Optimization Algorithm, the internal parameters of the control strategy are effectively tuned, enabling a control solution that achieves both high performance and robustness. Chen et al. [54] enhanced silicon wafer dicing accuracy by designing a high-stability servo control system combining high-definition closed-loop architecture with a PID + feedforward + notch filter algorithm. The dual-strategy approach optimizes both control parameters and dicing performance.(2)$$\;CMDout\left(n\right)=2^{-19}\times Ixx30\times \left[\begin{array}{c} \left\{Ixx08\times \left[FE\left(n\right)+\frac{Ixx33\times IE\left(n\right)}{2^{23}}+\frac{Ixx32\times CV\left(n\right)+Ixx35\times CA\left(n\right)}{128}\right]\right\}\\ -\frac{Ixx31\times Ixx09\times AV\left(n\right)}{128} \end{array}\right]$$

Notes: *Ixx*30: Proportional gain; *Ixx*31: Differential gain; *Ixx*32: Speed feedforward gain; *Ixx*33: Integral approach; *Ixx*35: Acceleration feedforward gain; *Ixx*08: Position register scale factor; *FE*(*n*): Following error; *IE*(*n*): Integral follow-up error; *CV*(*n*): Command speed; *CA*(*n*): Commanded acceleration; *AV*(*n*): Actual speed.

**Figure 9 micromachines-16-01188-f009:**
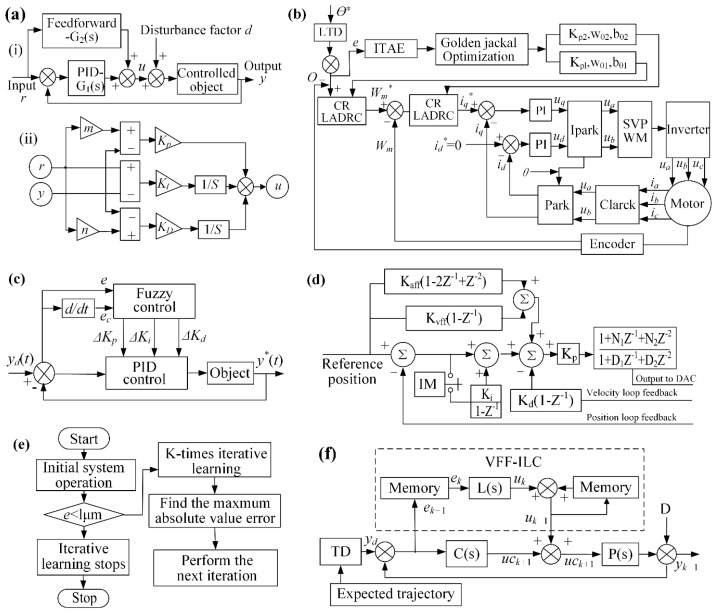
The servo control methods: (**a**) PID control principle and block diagram [52]: (i) Principle of PID+Feedforward Two Degree of Freedom Control; (ii) Block Diagram. (**b**) Parameter-optimized GJO algorithm [53]. (**c**) PID feedforward filter control [55]. (**d**–**f**) Fuzzy control and VFF-IL control [56].

The schematic of the PID+ velocity/acceleration feedforward+ Notch filter control algorithm is illustrated in Figure 9d. Shi et al. [55] proposed a Variable Forgetting Factor Fuzzy Iterative Learning Control (VFF-FILC) with tracking differentiator to address positioning accuracy issues in dicing paths caused by *Y*-axis parallelism. By integrating fuzzy control with iterative learning, this method effectively mitigates limitations of conventional PID control. Compared to traditional PID, VFF-FILC achieves a 28.57% reduction in learning time and demonstrates superior *Y*-axis precision control. Cao et al. [56] proposed a fuzzy controller that handles system nonlinearities and uncertainties by incorporating nonlinear, rule-based control logic, as illustrated in (Figure 9c,e,f). Controller parameters were optimized through heuristic algorithms including Particle Swarm Optimization and Genetic Algorithms for quantization and scaling factor calibration, ultimately yielding an optimal control strategy. Experimental results confirm an 11.8% reduction in control error and 26% improvement in tracking performance. Maharoof, M et al. [57] addressed the servo control of a positioning table by incorporating a disturbance observer and state estimator, which precisely estimates input disturbance forces to compensate for high-frequency spindle speed harmonics during dicing, thus minimizing spindle vibration position error. Compared to its predecessor, the latter controller reduced the positional Root Mean Square Error (RMSE) by approximately 92% to 96% in both numerical simulations and experiments, thereby establishing a new benchmark for control system accuracy.

Achieving dicing precision in servo systems depends critically on selecting the right control method, as shown in Table 3. The dicing saw’s *Y*-axis servo system, which employs fuzzy iterative learning control with tracking differentiators, demonstrates enhanced accuracy and control by effectively mitigating noise interference. Conversely, *X*-axis control minimizes micro-errors in high-speed spindle rotation to improve anti-interference performance, with simulations confirming the reliability of both approaches. Wang [58] designed a data-driven neural network iterative control method that progressively reduces tracking errors and enhances repetitive positioning accuracy through continuous control input refinement.

## 5. Key Component Optimization

With the development of the semiconductor industry, precision grinding wheel dicers are constantly becoming more intelligent and precise. Research efforts have been directed toward advancing control systems while concurrently optimizing the structural design of grinding wheel spindles and their key components.

### 5.1. Spindle Precision

The spindle drive unit comprises a DC spindle motor, motor shaft, and main spindle, utilizing armature signals to command the DC motor for spindle rotation. Xu et al. [59], working with the ADT7100 dicing machine, developed solutions for worktable flatness errors, cutterhead end-face runout, and misaligned end-face/*X*-axis parallelism. The workbench must maintain a flatness tolerance of 10 µm, as exceeding this limit will result in inconsistent cutting quality and increased contamination of the wheel bond and wafer surface, necessitating regular cleaning and reinstallation during operation. Simultaneously, the cutter head’s end face runout and its parallelism to the *X*-axis must not exceed 1 µm over 40 mm, since deviation beyond this range may cause uneven scratches, significant unilateral edge chipping, or even blade fracture during the dicing process, as shown in Figure 10a,b. Three blade alignment conditions are presented, ranging from a perpendicular blade surface with co-planar trajectory, to a non-perpendicular surface causing trajectory deviation, and finally to a perpendicular surface misaligned with the workpiece motion direction, as shown in Figure 10b. Spindle vibration errors critically impact dicing precision, primarily stemming from poor dynamic balancing, excessive vibration, processing overloads, and improper cutterhead installation. These issues can induce excessive tool stress during operation, leading to workpiece chipping and surface pitting defects. Therefore, Wang [60] rigorously analyzed common air-bearing spindle anomalies and proposed mitigation methodologies. In production practice, common issues in air bearing spindle cutting include spindle seizure, accuracy loss, tool vibration, and step errors. Spindle seizure stems from air supply contamination or pressure loss, requiring shaft and bearing repair. Accuracy issues arise from spindle vibration or improper tool installation, necessitating strict environmental controls and standardized operational procedures.

### 5.2. Tool Wear Height Measurement

Tool wear is an inherent phenomenon in grinding wheel dicing processes and a primary factor affecting dicing quality. Accurate measurement of blade wear and real-time compensation of dicing feed are therefore essential to maintain consistent dicing depth and ensure optimal dicing results. Contact height measurement utilizes the electrical circuit formed during instantaneous blade-worktable contact as its trigger signal. In contrast, non-contact measurement triggers when the blade edge interrupts the optical path of an opposing fiber sensor. The contact method serves as a fundamental technique mainly used for calibration purposes, whereas the non-contact approach represents a versatile solution widely implemented in production environments. Therefore, Hao et al. [61] investigated both contact and non-contact height measurement methods for grinding wheel dicing, as illustrated in Figure 10c(i,ii). Their study analyzed the operational principles of both techniques, quantified the effect of assembly deviations on non-contact measurement accuracy, and presented comparative deviation rates in Figure 10c(iii).(3)$$\Delta R^{\prime }=\sqrt{R^{2}-\Delta X^{2}}-\sqrt{{\left(R-\Delta R\right)}^{2}-\Delta X^{2}}$$(4)$$\;\Delta P=\Delta R^{\prime }-\Delta R$$(5)$$\;\gamma =\frac{\Delta P}{\Delta R}\times 100\%$$

Here, *R* denotes the pre-wear blade radius; Δ*X*, the horizontal deviation of the optical fiber detection point; Δ*R*, the actual wear depth; Δ*R*′, the calculated wear from non-contact height measurement; and Δ*P*, the non-contact measurement deviation. Experimental measurements from the DFD6362 dual-axis automatic wafer dicer show that Δ*P* progressively decreases with increasing blade radius when Δ*X* is held constant. Contact height measurements exhibit 4 μm repeatability, while non-contact methods achieve 3 μm repeatability. At Δ*X* ≤ 5 mm, the theoretical deviation Δ*P* = 0.2244 μm constitutes less than 8% of the measured accuracy (3 μm), indicating negligible impact of horizontal installation errors on non-contact measurement precision under these conditions.

### 5.3. Grinding Wheel Structural Optimization

Strategic structural optimization of grinding wheel dicing modules significantly improves dicing accuracy. In contrast, conventional wafer mounting methods frequently provide inadequate fixation, leading to tool instability and edge chipping. Amri et al. [62] addressed this challenge by developing advanced wafer fixation technology and examining chip formation and vibration behavior in double-side half-sandwich and full-sandwich wafer configurations. Compared to conventional single-sided mounting, the new wafer fixation technology provides superior wafer retention, reduced vibration, and improved dicing performance. Furthermore, Li et al. [63] investigated the impact of all parameters of the grinding wheel’s angle module on wafer dicing quality. Structural modifications to the angle module addressed inherent deficiencies in the original design, as shown in Figure 10d. The upgraded configuration improves suction cup leveling, enhances wheel positioning accuracy, and reduces wafer defects. Nonetheless, excessive frictional heat and inadequate heat dissipation cause thermal deformation and edge chipping, making efficient cooling essential for dicing quality and dimensional stability.

In industrial dicing, deionized water is widely used for cooling and debris removal, with studies confirming its superior cooling performance. Wang et al. [64] comprehensively detailed grinding wheel dicing system principles, cooling shroud functionalities, structural design methodologies, and critical component specifications to ensure optimal thermal management. By implementing a high-vacuum wafer cooling system with ionic liquids, Okabe et al. [65] achieved micrometer-scale dicing precision in experiments, surpassing nitrogen cooling limitations through superior thermal control. Concurrently, Yin et al. [66] engineered a fully automated dual-axis precision dicer featuring a gantry architecture, as shown in Figure 10e, with parallel-configured Y1/Y2 spindles, full closed-loop control, and DD motor-driven rotary table. Dicing tests revealed 5.2 μm maximum groove depth deviation and 4.8 μm positional error, producing surfaces with <5 μm edge chipping and burr-free morphology.

The developed dual-axis dicing system achieves an 80% higher efficiency and significantly superior cutting quality compared to conventional domestic single-axis machines.

**Figure 10 micromachines-16-01188-f010:**
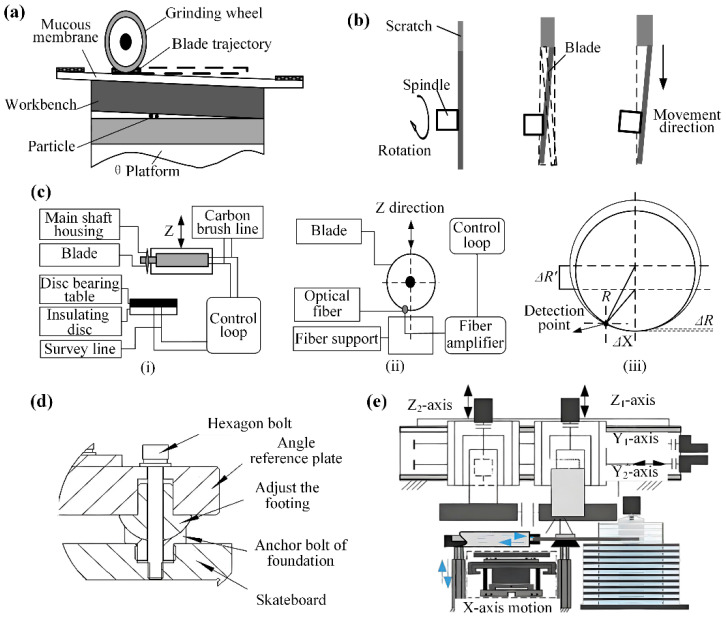
Schematic diagrams of key mechanical system characteristics: (**a**) Spindle system errors [59]. (**b**) Relationship between tool face and axes motion [59]. (**c**): (i) Principle of contact height measurement Contact [61]; (ii) Principle of non contact height measurement [61]. (iii) Non contact height measurement deviation calculation. (**d**) Angular module leveling structure [63]. (**e**) Dual-axis scribing machine structure [66].

## 6. Experimental Analysis

Ultra-thin diamond grinding wheel dicing is a high-precision process employed in semiconductor, electronic, and optical manufacturing, with its dicing quality being critical to subsequent production stages. Beyond numerous process parameters, key determinants of workpiece material removal rate, surface roughness, and grinding wheel blade wear status are illustrated in Figure 11.

### 6.1. Material Removal Mechanism and Simulation

During the high-speed rotation of the grinding wheel, its abrasive particles continuously rub, extrude, and cut the silicon wafer surface. In actual production, workers typically segment larger silicon wafers by guiding a dicing wheel along predefined paths at a set feed speed. This process removes material in a controlled manner, generating precise chips.

A well-defined dicing model framework facilitates parameter optimization and accurate prediction of tool stress, deformation, and wear. Consequently, condition-specific models are commonly developed to improve material removal rates in grinding wheel dicing. Wei et al. [67] developed an irregular three-dimensional particle model through micro-to-macro analysis of individual diamond abrasive grains. By incorporating grain spacing and spatial distribution, they constructed and analyzed a comprehensive simulated 3D morphology of the diamond grinding tool’s end face. This work provides a theoretical foundation for understanding the material removal mechanism in subsequent diamond blade cutting processes. However, Li et al. [68] investigated abrasive grain degradation mechanisms through analytical kinetic simulations and single-grain dicing testing. Conventional diamond abrasives exhibited predominant wear modes of surface disintegration and bulk fracture, whereas multi-edge diamonds demonstrated micro-fracture and cleavage-dominated wear, as shown in Figure 12a. The pronounced “multi-edge” effect during wear progression accentuates ductile removal characteristics in 4H-SiC materials. Zhang MH et al. [69] employed stochastic algorithms to reconstruct grinding wheel topography and identify active grains. They established a microstructure-aware cutting force model that incorporates removal mode transitions, achieving precise dynamic force prediction. This work provides theoretical insights for analyzing surface formation and defects in C/SiC grinding. Herbrandt et al. [70] developed a simulation model for single-diamond grain cutting that rapidly predicts grinding forces and material removal. Using optimization algorithms to balance efficiency and realism, it provides an effective tool for process optimization and prediction. Egea AJS et al. [71] found that the wear of diamond saw blades in dry concrete cutting can be evaluated by changes in outer diameter and weight. Dicing efficiency is not solely determined by the material removal rate, but is more significantly influenced by diamond concentration, matrix contact area, and bonding strength. High diamond density and strong adhesion improve wear resistance, whereas significant diamond detachment leads to a sharp decline in performance. Experimental research by Araujo et al. [72] revealed that the material removal mechanism for high-alumina ceramics is primarily inter/trans-granular fracture at the grain level, yielding small chipping and high-quality edges; for monocrystalline silicon; however, removal relies on elastoplastic deformation and macroscopic fracture, resulting in comparatively larger chipping. Gao et al. [73] established a finite element model for wire-sawing monocrystalline SiC containing spherical voids in Figure 12b. Numerical simulations revealed significant stress concentration variations depending on defect size/spatial relationships during dicing. Li et al. [74] demonstrated pseudo-plastic removal of single-crystal SiC by maintaining undeformed chip thickness below the critical depth 17.8 nm using coarse abrasives at low concentration. By substantially reducing chip thickness, this method effectively suppresses brittle fracture and edge chipping, enables efficient material removal, and preserves surface integrity and functionality. Kundrák et al. [75] combined theoretical and finite element models to analyze six bond characteristics’ effects on stress during diamond grinding of hard-brittle materials, finding that high-modulus bonds enhance grit adhesion, improve efficiency, and are better for high-speed conditions.

Riaz et al. [76] applied a constitutive material model to study dicing mechanisms in CFRP composites. Their results show that carbon fibers primarily fail via flexural-shear at deeper cuts, in contrast to the pure shear failure observed at shallower depths. Liu et al. [77] elucidated the dual-interface dicing mechanism for PZT-4H ceramics in Figure 12c, where the deburring wheel’s cylindrical surface removes material while its lateral face simultaneously polishes the kerf to achieve superior surface finish. Yuan et al. [78] developed an innovative high-speed micro-dicing technique for carbon fiber reinforced polymer (CFRP) composites. This technique enables a detailed analysis of the dicing mechanism, revealing that the dicing behavior varies distinctly with fiber orientation, as shown in Figure 12d. Implementation with diamond abrasive grains at an approximately 7 μm dicing depth produces sufficiently minimized dicing forces to prevent macroscopic fiber bundle deformation or fracture. Furthermore, the measured deformation resistance varies significantly with specific fiber orientation patterns. In controlled single-scratch tests, Duan et al. [79] used <111> and <110> diamond grains on the C-face and Si-face of single-crystal SiC, respectively. C-face dicing promoted micro-edge formation, and multi-edge interactions reduced groove-side fracture occurrence and size through damage interference. Zahedi, A. et al. [80] applied single-grit indentation fracture mechanics to model ceramic grinding as micro-scale interactions between random grains and the workpiece. This approach reveals the material removal mechanism by accounting for grit size, geometry, and material properties, offering a comprehensive analysis. Concurrently, Wang et al. [81] confirmed electroplated diamond dressing wheels with 213 μm grains exhibit optimal wear resistance while maintaining sharp edges and high protrusion post-wear.

Developing accurate dicing force models and understanding material removal mechanisms are essential for parameter optimization, tool selection, and efficient dicing operations. These models conserve material, reduce waste, and are fundamental to precision machining.

**Figure 12 micromachines-16-01188-f012:**
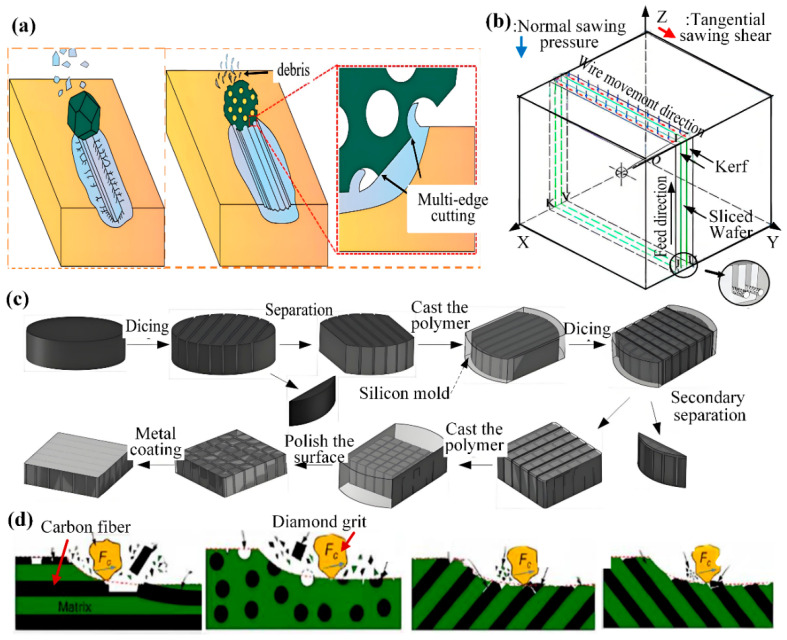
Schematic diagram of material removal mechanisms: (**a**) Diamond abrasive particle removal [68]. (**b**) SiC model partitioning [73]. (**c**) Piezoelectric composite removal [77]. (**d**) Removal at different carbon fiber orientations [78].

### 6.2. Process Parameter Influence and Optimization

Enhancing workpiece machining performance has driven research attention toward ultra-thin diamond blade fabrication and dicing behavior, with process parameter settings exerting a significant influence on dicing performance. In multi-wire sapphire slicing, Liu et al. [82] identified the normal cutting force and its variation (ΔF_n_) as key stability indicators. Their work shows that parameters like wire speed (V_s_), swing angle (θ_max_), wire consumption (M_d_), tension (F_w_), and dicing (T) each distinctly affect ΔF_n_, whose fluctuation is directly tied to wire saw wear, enabling optimized process stability. However, Agudon et al. [83] compared the edge chipping coefficient of variation (COV) between high and low feed rates, finding that the lower feed rate resulted in an improved COV, which helps minimize edge chipping. Wang et al. [84] proposed high-speed micro-dicing of SiC using two diamond blade types: resin-bonded and metal-bonded. For silicon carbide dicing, the optimal parameters using a resin-bonded blade are a spindle speed of 20,000 rpm, a feed rate of 4 mm/s, and a dicing depth of 0.1 mm.

Due to the superior performance of step cut mode in reducing scratch damage on wafer surfaces, Su et al. [85] applied this method to create 60 μm trenches during experimentation. The research team investigated the effects of spindle speed and dicing depth on edge chipping via two-way ANOVA, determining the optimal parameters through analysis of marginal means and prediction plots. These optimized parameters, specifically 50,000 rpm spindle speed, 50 μm dicing depth, and 30 mm/s feed rate, achieved substantial decreases in surface edge chipping. Shi et al. [86] designed orthogonal array experiments investigating process parameters including spindle speed, feed rate, dicing depth, blade cooling water flow, and spray water flow. The team then constructed a regression model and performed iterative optimization, comparing the performance of Genetic Algorithm (GA) and Particle Swarm Optimization (PSO) methods. Figure 13a demonstrates the influence of each parameter on maximum edge chipping size. GA optimization yielded a maximum edge chipping width of 38.54 μm, corresponding to an 8.23% reduction compared to conventional methods while meeting performance requirements. Ma et al. [87] employed orthogonal experimental design and robust design methodology to analyze data obtained from cross dice width tests and maximum cross dice width tests. Experimental results demonstrate that maintaining the dicing width below 40 μm, with a maximum under 65 μm, significantly enhances dicing quality. She et al. [88] investigated key characteristic parameters of the dicing blade, such as abrasive grain size and abrasive concentration, for dicing Al_2_O_3_ ceramic substrates, as shown in Figure 13b. Utilizing orthogonal experimental design and a Gaussian Process Regression (GPR) model for modeling and algorithm optimization, the optimal results differed by only 0.5 μm from experimental validation results. Wang et al. [89] combined simulation modeling and experimental simulations to study wafer-to-wafer bonding warpage, finding a 27.8% reduction when the pitch size was decreased to one quarter of its original value, in agreement with simulation results. Amrita Chaudhary Li et al. [90] found traditional simulation stealth dicing simulation time-consuming and costly. They proposed a digital twin and ML-based system that uses XGBoost to learn the complex relationship between process parameters and the stress intensity factor (SIF) from the simulation model. Li et al. [91] obtained a series of dynamic blade diameters by varying the rotational speed, static diameter, and elastic modulus of the diamond dicing blade in Figure 13c. This method establishes a foundation for tool diameter compensation in high-speed precision dicing. Using a 3D finite element simulation model, Tan et al. [92] found that thinner silicon dicing pieces exhibit a wider range of fragment rates, indicating reduced fracture strength. Araujo et al. tested diamond wheel cutting of high-purity (99.8%) alumina, varying depth (1–3 mm), feed (1–19 mm/s), and speed (10,000–30,000 rpm). At 30,000 rpm, the wheel’s rigidity ensured stable dicing [72]. Yang et al. [93] developed an Abaqus model to simulate micro-cutting of silicon carbide with a conical diamond abrasive grain. Pre-simulation set the maximum dicing depth at 1.50 μm, with optimal parameters identified as a depth of 0.50 μm, a speed of 76 m/s, and an edge angle of 60°. Results show that dicing depth is the dominant factor affecting dicing force. Increasing the dicing speed appropriately also improves efficiency while maintaining quality. Hertz contact theory analysis confirmed that the loading force positively correlates with friction force, dicing force, and indentation depth, ultimately producing a smooth and flat cut edge morphology.

Given the significant influence of spindle speed, feed rate, and dicing depth on dicing quality, a systematic analysis of their interactions is imperative for determining the optimal combination to achieve high-efficiency, high-quality dicing.

## 7. Prospects and Recommendations

Diamond grinding wheels are essential in modern high-precision machining, with extensive applications in the semiconductor, integrated circuit (IC), and aerospace industries. Ongoing technological advances persistently increase the demand for superior dicing accuracy. While studies on diamond grinding wheel dicing machines have achieved notable progress, persistent technical challenges remain, calling for broader collaborative research efforts.

Rational Design of Dicing Processes. High-quality dicing processes minimize edge chipping and preserve workpiece integrity during grinding wheel operations. While single-process methods have inherent limitations, multi-mode and composite dicing techniques combine complementary advantages to reduce dicing-induced damage. Current diamond grinding wheel dicing predominantly employs a wheel-laser hybrid approach, which not only overcomes laser depth inaccuracies but also facilitates precision machining of complex geometries. Future integration of wheel dicing with plasma cutting and electrolytic systems, combined with optimized process design, will streamline operations and improve workpiece quality.Process parameter optimization is evolving from singular core parameter adjustments into a comprehensive systems engineering approach that integrates tool characteristics, workpiece properties, and cutting mechanisms. Next-generation intelligent dicing systems will leverage advanced algorithms to develop digital models that capture the complex interrelationships among abrasive composition, bond matrix properties, composite workpiece behavior, and material removal physics. Leveraging this foundational understanding, these systems will perform intelligent decision-making through dynamic analysis and synthesis of multidimensional factors, autonomously generating optimal parameter combinations tailored to specific processing conditions. This paradigm shift fundamentally advances cutting precision, surface integrity, and process consistency to unprecedented levels.Dual improvement of cutting accuracy and efficiency. The growing demand for dicing precision in photoelectric semiconductors is driving diamond grinding wheel processes toward higher accuracy and efficiency. By reasonably designing the structure of grinding wheel, the shape and thickness of dicing edge and the working parameters of dicing, ultra-thin cutting of the workpiece can be achieved. This enables the formation of narrower kerfs and suppresses edge chipping, leading to higher levels of dicing accuracy and efficiency. From a materials perspective, significant advances can be made by developing novel bond materials and innovative abrasive grains. A synergistic design with advanced bond systems facilitates control of the material removal process at the atomic scale. This approach enhances the grinding wheel’s strength and toughness, while adjustments to the formulation ratio regulate diamond abrasive concentration, thereby improving the wheel’s overall toughness and wear resistance.Develop a system simulation model for sustainable and environmentally friendly operations. Improper parameter settings by operators during diamond wheel dicing can cause system failures and reduce process efficiency. Integrating a real-time virtual dicing simulation model into the dicing machine allows for preemptive identification and correction of such issues. By simulating the process in advance, optimal parameters can be established, thereby reducing material loss and enhancing dicing stability and reliability. Furthermore, in wheel dicing operations, precise temperature control and extended wheel service life remain critical customer-valued performance factors, necessitating an efficient and rational cooling and waterproofing system. This can be accomplished through several approaches: configuring periodic coolant filtration; deploying high-velocity spray pipes or vortex-tube bidirectional drying for the worktable and blades or optimizing coolant formulations to improve cooling performance.Efficient combination of intelligence and automation. Advances in modern technology, especially in networking and data processing, are creating new opportunities for photoelectric material dicing. Intelligent and automated system software can be integrated to apply deep learning. This would enable real-time monitoring of abnormal conditions in the grinding machine’s operation. Such a system could automatically adjust dicing parameters to optimal levels, monitor grinding wheel wear, and compensate for it. This allows for the timely correction of improper operations, thereby reducing damage and scrap.

## Figures and Tables

**Figure 1 micromachines-16-01188-f001:**
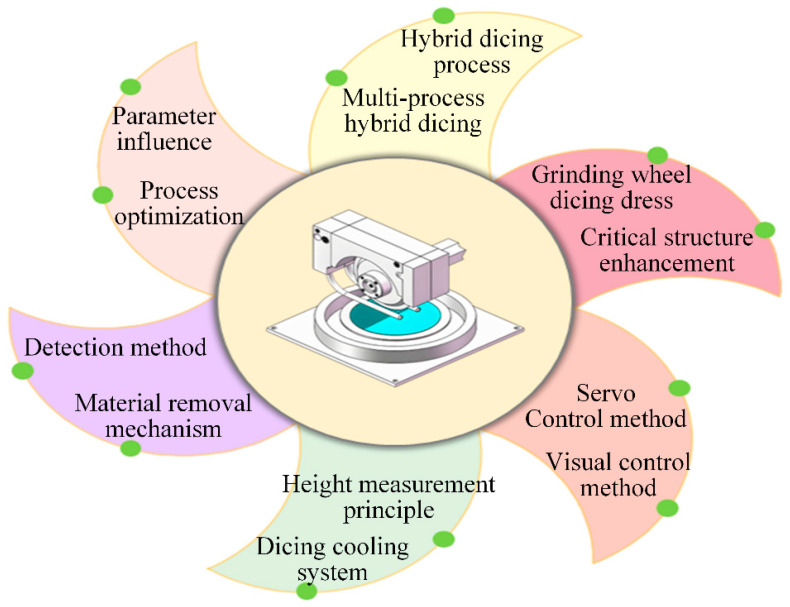
Schematic diagram of the grinding wheel dicing process.

**Figure 2 micromachines-16-01188-f002:**
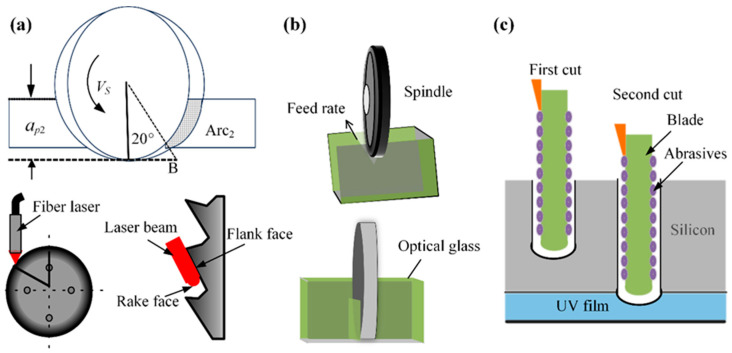
Three methods for machining polymorphic dicing: (**a**) Alternating contact arc and laser scratches [4,5]. (**b**) Double grinding wheel dicing [6]. (**c**) Layered dicing [7].

**Figure 3 micromachines-16-01188-f003:**
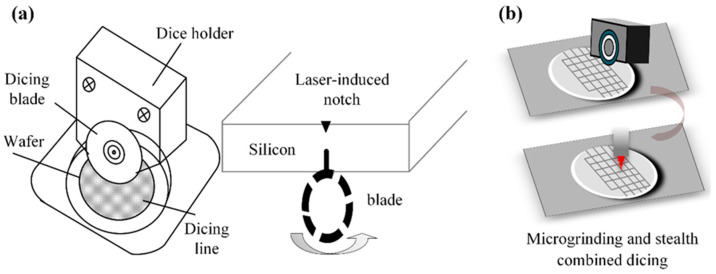
The compound dicing processing techniques, including: (**a**) Laser wheel compound dicing [8]. (**b**) Micro-grinding stealth dicing [10].

**Figure 5 micromachines-16-01188-f005:**
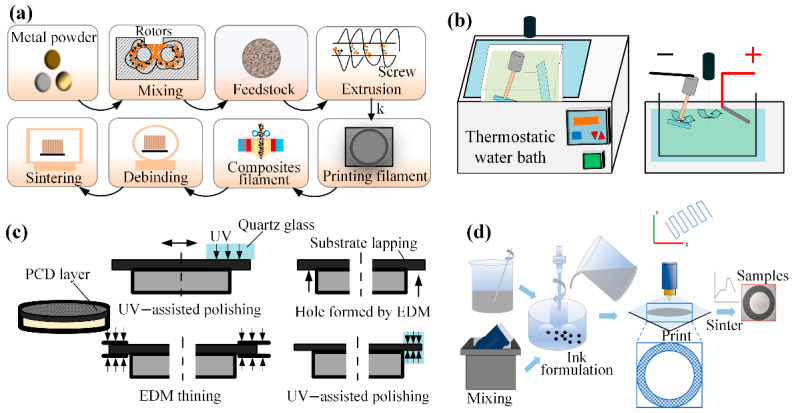
Multi-process preparation techniques: (**a**) Molten wire manufacturing method [19]. (**b**) Composite electrodeposition technology [20]. (**c**) PCD blade fabrication [21]. (**d**) Ceramic ink preparation [25].

**Figure 11 micromachines-16-01188-f011:**
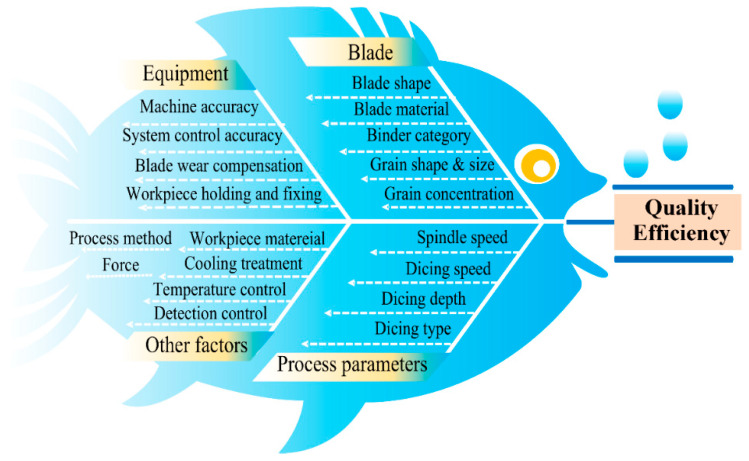
Fishbone diagram of factors influencing dicing surface quality.

**Figure 13 micromachines-16-01188-f013:**
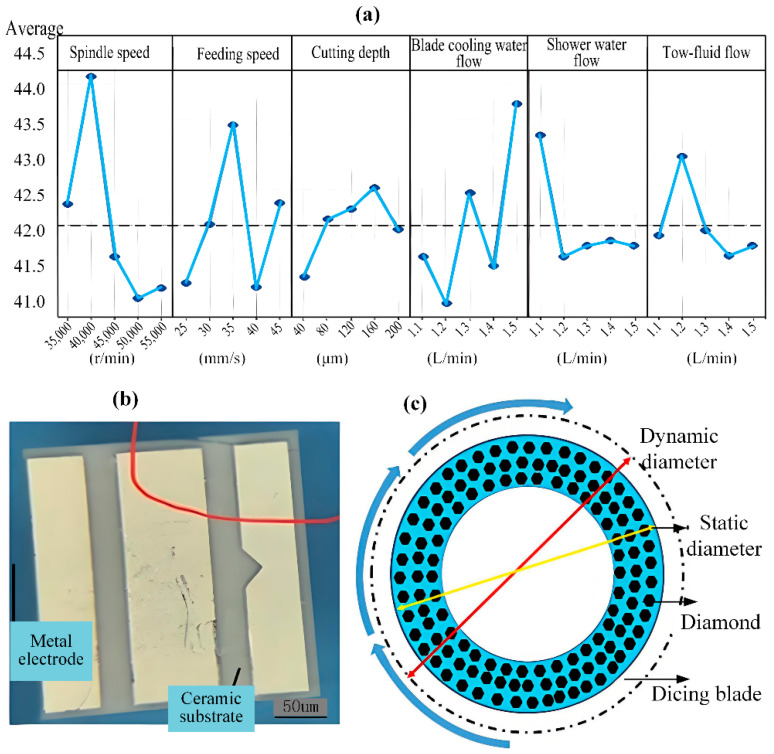
Key machining characteristics: (**a**) Effects of multiple factors on maximum dicing chip width [86]. (**b**) Flank face edge collapse [88]. (**c**) Dynamic diameter variation diagram [91].

**Table 1 micromachines-16-01188-t001:** Methods for predicting tool wear and dicing quality.

Predictive Model	Application Advantages	Limitations
AGJO-GRU Predictive model	Rapid convergence and high predictive accuracy	None
AGA-BPNN Predictive model	Powerful global search ability, ensuring reliable avoidance of local optima	None
CLSSA-BILSTM	High prediction accuracy and smaller error	Systems with numerous variables are prone to getting trapped in local optima.
BPNN monitor	Real-time monitoring of chip conditions and buffering of chip accumulation	Unstable training process and convergence to local optima
Edge diffraction-based monitoring	Non contact detection, high efficiency, strong anti-interference ability	High requirements for the work environment and large equipment investment
Fiber optic sensor detection	High precision, precise wear monitoring	Measurement during contact may result in tool damage
Bidirectional Attentive Temporal ResNet with Voting	Accuracy > 95%, minimizing equipment false alarms and shutdowns to the greatest extent possible	High computational complexity and dependence on device computing power requirements

**Table 2 micromachines-16-01188-t002:** Comparison of Visual Control Methods.

Control Algorithm	Advantages	Disadvantages
Based on nonlinear image enhancement algorithm and template matching algorithm	Image preprocessing, improving the robustness of localization, and high fitting accuracy	Unable to convert physical size accuracy, limiting camera usage
Air bearing technology + CCD sensor	Completely non-contact measurement, avoiding physical damage to equipment workpieces	The system is complex, requiring high precision design for air bearing, and the cost is high
Image pyramid + improved template matching based on grayscale and line characteristics	Fast search speed, strong robustness, suitable for fast-paced production lines	Poor applicability
Bilinear interpolation and sub-pixel edge detection algorithms	Sub-pixel technology has high edge positioning accuracy and halcon platform has trajectory planning capability	Commercial software licensing issues, specific indicators not quantified
Optimization of sensor measurement points	Strong perception accuracy and reliability	Dependent on visual positioning system

**Table 3 micromachines-16-01188-t003:** Comparison of Servo Control Methods.

Control Algorithm	Advantage	Disadvantage
PID + feedforward 2DOF control algorithm	Clear structure, inherent stability, and a balanced dynamic-static performance	Limited suppression of nonlinear/time-varying disturbances
Cascaded reduced-order LADRC tuned by Gold Jackal Optimization	High robustness with automated parameter optimization	High complexity in both theory and computation
PID+ velocity/acceleration feedforward+ Notch filter control algorithm	Strong practicality and can improve the accuracy of the cutting machine	More accurate system identification models are needed
Variable forgetting factor fuzzy iterative learning control (VFF-FILC) with tracking differentiator	Highly suitable for repetitive tasks, with strong disturbance rejection and fast convergence	The algorithm is complex and suitable for repetitive motion trajectories
Fuzzy Control + Heuristic Algorithm (PSO/GA) Optimization	Effective for model-free nonlinear systems	The optimization process is complex
Data-driven neural network iterative control	Strong nonlinear fitting force	Requires a large amount of data for training

## Data Availability

No new data were created or analyzed in this study.

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
