# Peer review of "Recent Advances in Precision Diamond Wheel Dicing Technology"

_micromachines, 2025, doi:10.3390/mi16101188_

Round 1
Reviewer 1 Report
Comments and Suggestions for Authors
This review paper summarizes the key dicing technologies of precision diamond grinding wheels, analyzing the advantages and disadvantages of composite and multi-mode dicing processes. It discusses the influence of blade preparation, equipment accuracy, and the optimized design of key components on dicing quality. The paper also introduces tool wear modeling, material removal mechanisms, and some methods for optimizing process parameters. It provides an outlook on precision diamond grinding wheel dicing technology, demonstrating certain engineering significance. The following issues need to be addressed to further improve the paper:
- The analysis of dicing simulation technology in the paper is not sufficiently in-depth. It is necessary to supplement relevant literature for a more systematic analysis.
- The structure of the paper requires slight adjustments. The section "5.2 Tool wear" should be repositioned before "2.3 Grinding wheel dicing trimming" to align with the main theme of Chapter 2.
- Although Chapter 3 cites numerous references to describe Vision system control and Servo control system in dicing processes, it lacks a summary analysis and comparison of various methods. Further elaboration is needed in this regard.
- The description and analysis of some figures in the paper are insufficient. For instance, in Chapter 4, the analysis of Figures 9a and 9b is inadequate and requires more detailed elaboration.
- Equation (2) lacks a description of the physical significance of its variables, which should be provided.
Author Response
Please check the PDF file, thank you.

Reviewer 2 Report
Comments and Suggestions for Authors
Precision dicing with diamond wheels is vital in semiconductor, IC, aerospace, and related fields for its high precision, efficiency, and wide material range. This review summarizes core principles, processes, and blade preparation, analyzing composite and multi-mode cutting to reduce defects versus single-mode methods. It surveys blade fabrication (e.g., metal-binder sintering, roll forming), machine-vision and servo algorithms for feature recognition and scribe-line control, and key grinder components to limit deviation. Models for tool wear, material-removal mechanisms, and the effects of spindle speed, feed rate, and depth on quality and kerf width are discussed. Finally, it outlines future directions for advancing precision dicing technologies. All references cited are Asian sources. Where is: On the cutting performance of segmented diamond blades when dry-cutting concrete, Materials 11 (2), 264
Figures are too complex. Figure 8 is difficult to read and understand.
Eliminate Fig 10. Aim ate optimization with no philosophical figures.
Conclsuions 2 is too difficult to describe.
Please define the soruce of abrasive grains, as in missed Works is done.
Removal mechanism is well described, congrats.
Why did you say: ultra-thin?
Paper needs changes.
Author Response
Please check the PDF file, thank you.
